# Intervention fidelity and factors affecting the process of implementing a mobile phone text messaging intervention among adolescents living with HIV: a convergent mixed-methods study in southern Ethiopia

Abayneh Tunje ![ORCID] ,[1,2] Helene Åvik Persson,[1] Degu Jerene,[3] Inger Hallstrom[1]

[1]Department of Health Sciences, Lund University Faculty of Medicine, Lund, Sweden
[2]Public Health, Arba Minch University, Arba Minch, Ethiopia
[3]KNCV Tuberculosis Foundation, Den Haag, The Netherlands

**Correspondence to**
Dr Abayneh Tunje;
abayneh_tunje.tanga@med.lu.se

## ABSTRACT

**Objective** To assess the intervention fidelity and explore contextual factors affecting the process of implementing a mobile phone text messaging intervention in improving adherence to and retention in care among adolescents living with HIV, their families and their healthcare providers in southern Ethiopia.

**Design** A convergent mixed-methods design guided by the process evaluation theoretical framework and the Reach, Effectiveness, Adoption, Implementation and Maintenance framework was used alongside a randomised controlled trial to examine the fidelity and explore the experiences of participants in the intervention.

**Setting** Six hospitals and five health centres provide HIV treatment and care to adolescents in five zones in southern Ethiopia.

**Participants** Adolescents (aged 10–19), their families and their healthcare providers.

**Intervention** Mobile phone text messages daily for 6 months or standard care (control).

**Results** 153 participants were enrolled in the process evaluation. Among the 153 enrolled in the intervention arm, 78 (49.02%) were male and 75 (43.8%) were female, respectively. The mean and SD age of the participants is 15 (0.21). The overall experiences of implementing the text messages reminder intervention were described as helpful in terms of treatment support for adherence but had room for improvement. During the study, 30 700 text messages were sent, and fidelity was high, with 99.4% successfully delivered text messages during the intervention. Barriers such as failed text messages delivery, limitations in phone ownership and technical limitations affected fidelity. Technical challenges can hinder maintenance, but a belief in the future of digital communication permeates the experiences of the text message reminders.

**Conclusions** Overall fidelity was high, and participants' overall experiences of mobile phone text messages were expressed as helpful. Contextual factors, such as local telecommunications networks and local electric power, as well as technical and individual factors must be considered when planning future interventions.

**Trial registration number** PACTR202107638293593.

## STRENGTHS AND LIMITATIONS OF THIS STUDY

⇒ Secure and confidential delivery of high-volume messages through a four-digit short code.
⇒ The use of a mixed-methods design and the inclusion of family members and healthcare providers is a strength of this process evaluation study.
⇒ There was a limited number of adolescents' family and healthcare providers included in the interviews.

## INTRODUCTION

Globally, 38.4 million people were living with HIV in 2021. Of these, 1.71 million were adolescents aged 10–19, 88% of whom live in sub-Saharan Africa.[1 2] To control the epidemic, the Joint United Nations Programme on HIV/AIDS launched the 95%–95%–95% targets aimed at HIV testing, treatment and viral suppression by 2030.[3] However, by 2021, 75%t of the 38.4 million people living with HIV were receiving antiretroviral therapy (ART), a treatment gap of 5.9 million people.[4] Moreover, achieving the recommended optimal adherence level of >95% which is required for ART to be effective, remains a challenge within the HIV care continuum.[5] This poses a significant threat to HIV interventions in resource-limited settings.[6]

Poor adherence to ART has several consequences, including increased risk of viral drug resistance and reduced treatment effectiveness towards viral suppression, leading to disease progression, greater risk of death and increased risk of viral transmission.[7] Some of the major barriers to ART adherence include socioeconomic status,[8 9] fear of being stigmatised or discriminated against as a result of one's HIV status,[10] forgetting to take

medication on time,[11] treatment fatigue[12] and patient–provider communication.[13] Several interventions to overcome these barriers have been studied in a variety of settings.[14] The use of digital health strategies to enhance HIV treatment is one of the priority interventions in sub-Saharan Africa.[15] Targeted digital client communication using mobile phone-based short text messaging has been recommended for adherence support for a range of health issues and adherence to antiretroviral treatment.[16] The success and effectiveness of an mHealth intervention in achieving desired health outcomes is dependent on the context in which they are implemented.[15 16] Context plays a crucial role in either the success or failure of an intervention.

In addition to design and implementation, the success of interventions is diverse and may be influenced by factors such as sociodemographic and culture.[17] Observing what is delivered in practice with close reference to the theory of the intervention can help evaluators in distinguishing the intervention fit in different contexts and changes that undermine intervention fidelity.[15 17] Trials should continue to rigorously assess outcomes but also include integral process evaluations that use qualitative and quantitative data to develop and test hypotheses about how interventions work.[18] In a multicentre trial, a process evaluation is necessary to understand whether the intervention was implemented and received similarly across all sites.[9] However, trials assessing mHealth adherence interventions for HIV often do not include process evaluations to examine the fidelity and quality of the intervention delivery, causal mechanisms for the health outcomes, contextual factors affecting the delivery and costs of implementation.[16 19] Understanding the entire implementation process of an mHealth intervention allows practitioners to interpret the results and replicate the intervention in other contexts.[20 21] Therefore, this study aimed to evaluate the process of a mobile text messaging intervention conducted among adolescents living with HIV in Ethiopia.

## METHODS
### Study design
A convergent mixed-methods design[17] combining quantitative and qualitative techniques was used. The design was guided by the process evaluation theoretical framework[22] and structured by the RE-AIM dimensions (Reach, Effectiveness, Adoption, Implementation and Maintenance)[18] to evaluate intervention effectiveness alongside a randomised controlled trial to examine the study's fidelity and to explore the experiences of the participants in the intervention.

The trial was registered in Pan African Clinical Trials Registry with the registration number PACTR202107638293593 on 17 July 2021.

### Study setting and participants
Six hospitals and five health centres that provide HIV treatment and care to adolescents in the Gamo, Gofa, Konso, South Omo and Wolayita zones in Southern Ethiopia were included in the main study. These zones are home to over 25 distinct ethnic groups, each with its own culture and context. Between 5 July 2022 and 28 February 2023, 306 adolescents living with HIV were included in the study, in either the intervention or in the control group. The quantitative data included process indicators while the qualitative interviews included ten adolescents from the main trial, four adolescents' families and two healthcare providers who were involved in the process evaluation. We used the following technique to identify eligible adolescents: first, we identified the number of adolescents of eligible age in each facility. Then, using their address, we called each adolescent and their parents or caregivers directly via phone numbers, care providers or adherence supports. Then, data collectors approached adolescents and posed literacy questions about their ability to read text by displaying sample texts. Then, eligible participants in the study were asked to give consent and assent. Thus, adolescents aged 10–19 years, diagnosed with HIV, currently participating in ART care, and planning to stay in the designated facility for at least 6 months following study enrolment were eligible. Participants who were above the age limit did not disclose their HIV status, were unable to read text or had a disability were all excluded. Finally, the data collector showed each eligible adolescent how to use their mobile device, including text saving and deletion.

### Theoretical framework of process evaluation
The RE-AIM framework[18] was used for the process evaluation to comprehensively examine the level of implementation. This involves understanding the implementation of interventions in terms of (1) fidelity, defined as to what extent the intervention was implemented consistently with the underlying theory as planned; (2) delivery, defined as to what extent all of the intended activities, training and materials were provided to programme participants; (3) reach, defined as the absolute number, proportion and representativeness of individuals who are willing to participate in a given initiative, intervention or programme; (4) receipt, defined as how participants reacted to specific aspects of the intervention and (5) context, defined as what contextual factors influence implementation or the intervention outcome.

### Description of the intervention
The MRC framework,[22] a systematic literature review,[23] interviews with adolescents[24] and the theoretical health belief model (HBM)[25] were used to develop the intervention. Adolescents living with HIV, healthcare providers and research project technical advisory team members participated in the design[26] under the guidance of a research advisory committee. The HBM is a theoretical model that has been constructed from six domains: susceptibility, severity, barriers, benefit, cues to action and self-efficacy. After the intervention's development, text messages were uploaded to the server, and an automated

message-pushing software was used to send tailored daily text messages using a four-digit code from Ethiopia Telecom. This code helps adolescents in avoiding messaging source-related stigma and distinguishing intervention messages from other messages.

Other various configurations such as wide-area network internet protocol (IP), local area network IP, Ethio telecom central server IP, a virtual private network line and office servers were used as infrastructure, as required to support the study. The text messages were categorised and delivered based on each adolescent's customised medication schedule and time. The software sent the message 15 min before each participant's medication time. The type of message was short and mainly focused on advice about the benefits of taking on time, the risk/consequences of missing/not taking on time, encouragement and without mentioning anything about HIV or the name of the drug. For example, 'Those who use it properly know its benefits', 'I'm taking it properly and on time'. For example, these messages in Amharic are 'በትክክል የሚወስዱት ጥቅሞቹን ያውቃሉ) and በትክክል እና በሰዓት አወስዳለሁ', respectively. The server was checked daily to troubleshoot network/cell phone issues. Based on the server information, the first author contacted the facility level care provider to contact the adolescent with corrective actions; for example, subscriber identity module card (SIM card) replacement, damaged mobile phones and network troubleshooting for maintenance and replacement.

## Sample and data collection

The quantitative data were collected from the server, activity logs, an adverse event monitoring form and an interview with adolescents. The researcher collects data on intervention fidelity by tracking the number of text messages sent, delivered and failed from the server daily.

Furthermore, each facility representative documented any adverse event related to the trial, filled the form and sent it to researchers every week. Furthermore, during the follow-up data collection phase, adolescents asked if they received text messages on a regular schedule.

A purposive sampling of 10 adolescents with HIV (aged 12–19), 4 family members/relatives (aged 36–45) and 2 healthcare providers (aged 31–36) from 5 different healthcare facilities were asked to be interviewed about their experiences in the intervention between 5 July 2022 and 28 February 2023, at healthcare facilities by the first author. The interviews were conducted in all respondent languages, such as Amharic, Wolaytigna, Gamogna, Gofigna, Arigna and Konsegna in a quiet and confidential area at the respective health facilities by the first author and one research assistant and lasted on average 30 min. All interviews were audio recorded with the permission of the participants.

An interview guide was used to gain a rich, in-depth understanding and to allow the participants to concretise their experiences of their use of text messages, communication and technical aspects of the intervention. The interview guide included subsections adapted

to the participants, that is, adolescents, families and healthcare providers (interview guide attached as online supplemental file). Hence, the questions were different, depending on to whom they were addressed. The introductory question to the adolescents was: 'Are you receiving the mobile text-message reminder from the hospital?', followed by follow-up questions such as, 'What do you think about the messages?' The families were initially asked 'How do family/social-related factors affect ART adherence among adolescents?' Healthcare providers were asked 'How did you experience communication with the adolescents with regard to the intervention?' Follow-up questions and prompts were used throughout the whole interview.

The interviews were audiotaped, transcribed verbatim and translated from Amharic to English by the first author and checked for accuracy by one independent language expert.

## Data analysis

The number of Short Message Service (SMS) messages sent and delivered to participants was tracked as a measure of intervention fidelity. When SMS messages did not arrive on the phone, the project advisory team investigated the external network and other infrastructure issues. If the message could not be delivered during this period, the computer automatically recorded it as a delivery failure and sent the message again. Data captured on the delivery status of the SMS messages were recorded as delivered (the phone had reception marked as green) and undelivered (the phone was switched off or marked as red in the server). The programme could not record whether messages delivered to the phones were opened.

The descriptive statistical analysis was used to summarise data in frequency, percentages, mean and SD. The quantitative effectiveness data is analysed separately and not included in the process evaluation. The RE-AIM outcomes framework components of Reach, Effectiveness, Adoption, Implementation and Maintenance were used to present mixed method findings about the delivery of a reminder message, success and intervention barriers for the intervention implementation process.[18]

Interviews were analysed by using the principles for content analysis described by Graneheim *et al*.[19 27] First, the transcripts were read several times to gain a sense of the whole by the first and the last author. Thereafter, meaning units were identified and condensed to reduce the text while maintaining the core meaning. The first author developed a coding frame for the analysis by using NVivo software, which was discussed by all authors. Condensed meaning units were then labelled with a code, which were kept close to the text on a manifest level. Next, subthemes were created from the code frame by the first and last authors. The subthemes were then abstracted into themes. Finally, the underlying meaning—the latent content—was described as a main theme. Thereafter, one of the authors (HÅP) read the transcripts, extracted meaning units, codes and themes.

The content was discussed and reflected on together with all authors and lasted until an agreement was reached.

## Patient and public involvement

Representatives of adolescents were involved in the development of messages for the SMS reminders and interview guides. Their feedback was used to adapt its contents and to set strategies to promote inclusivity in study participation during participant recruitment. In addition, participants, healthcare providers and the participants' families were involved in reporting any incidences related to mobile devices and other issues.

## RESULTS

The characteristics of adolescents assigned in the intervention arm are shown in table 1.

The results are structured using the RE-AIM framework[18] and corresponding themes (table 2).

## Reach

The total number of individuals that were assessed for eligibility was 435, of which 306 (70.34 %) meet eligibility criteria. The remaining 129 participants excluded for the following reasons: 27 (20.9%) were unable to stay in the study area for the duration of the study, 39 (30.2%) were unable to read, 24 (18.6%) were did not know their status, 21 (16.3%) were not reached, 13 (10.1%) refused to participate and 5 (3.9%) had an impairment. Thus, 306 participants were enrolled and randomised: n=153 in the intervention, and n=153 in the control group. Among the 153 enrolled in intervention arm, 78 (49.02%) were male and 75 (43.8%) were female, respectively. The mean and SD age of the participants is 15 (0.21). Of these, 111 (72.55%) resided in an urban area, 97 (63.40%) had primary school level education and 10 (6.54%) live alone (orphan).

## Effectiveness

The overall rate of retention at 6 months in the intervention group was 152/153 (99.4%) while in the control group was 151/153 (98.69%).

### Text message reminders were helpful for adherence but there was room for improvement

The overall experience of the intervention among adolescents living with HIV was that text message reminders were helpful when it came to their adherence to medication, but direct access to phones and better timing of messages should be assured to increase their functioning and usability. The adherence was improved and was the best achieved since the intervention was introduced. After 6 months of follow-up, the intervention group had a higher rate (114/152, 75%) than the control group (77/151, 51%). Despite this, more effort was requested concerning technical issues to make the reminders even more helpful. The participants described facilitating aspects like increased medication intake and communication. Also, the well-being of the adolescents and the

support function were highlighted. However, technical problems like lost telephones or text message reminders arriving after medication time were emphasised as barriers.

**Table 1** Participant characteristics

| Characteristics of adolescents assigned in the intervention | |
|---|---|
| **Characteristics** | **Frequency/mean (SD)/per cent** |
| Adolescent age, mean (SD) | 15 (0.21) years |
| Adolescents age category | |
| 10–14 | 60 (39.22) |
| 15–19 | 93 (60.78) |
| Adolescents gender | |
| Male | 78 (49.02) |
| Female | 75 (43.8) |
| Languages of adolescent during interview | |
| Amharic | 62 (40.52) |
| Wolaytigna | 27 (17.65) |
| Gamogna | 39 (25.49) |
| Gofigna | 13 (8.5) |
| Arigna | 6 (3.3) |
| Konsegna | 6 (3.3) |
| Educational status of adolescent | |
| Grades 1–8 | 97 (63.40) |
| Grade 9 and above | 56 (36.60) |
| Residence of adolescent | |
| Urban | 111 (72.55) |
| rural | 42 (27.45) |
| Living arrangement | |
| Live alone | 10 (6.54) |
| Live with family | 143 (93.46) |
| Psychosocial peer-support | |
| Participate | 47 (30.72) |
| Do not participate | 106 (69.28) |
| Interviewee characteristics | |
| Adolescents (sex/age) | |
| Male (10–14 years) | 2 |
| Female (10–14 years) | 3 |
| Male (15–19 years) | 2 |
| Female (15–19 years) | 3 |
| Adolescents level of education | |
| Grades 1–8 | 5 |
| Grade 9 and above | 5 |
| healthcare providers (sex) | |
| Male | 1 |
| Female | 1 |
| Parent/caregiver (sex) | |
| Non-biological caregiver | 2 (1 male and 1 female) |
| Biological | 2 (1 mother and 1 father) |

**Table 2**  Overview of the qualitative themes related to the dimensions in the RE-AIM framework

| RE-AIM dimension[18] | Definition[18] | Qualitative theme |
|---|---|---|
| Reach | The number, proportion and representativeness of eligible individuals who participate in each initiative. | |
| Effectiveness | The impact of an intervention on the relevant outcomes, including potential adverse effects, quality of life and economic outcomes. | Text message reminders were helpful for adherence but there was room for improvement |
| | | Text message reminders support adherence |
| | | Feeling comfortable receiving text message reminders |
| | | The code delivered with text message reminders minimises the risk of stigma. |
| Adoption | The reach and effectiveness/ efficacy of an intervention at the setting level. | Supporting the text message reminders |
| Implementation | The intervention agents' fidelity to the various elements of an intervention's protocol, including consistency of intended delivery. | Failing text messages delivery influences medication intake |
| | | Limitations in phone ownership |
| | | Technical limitations |
| Maintenance | Connects with both setting-level indicators (the extent to which a programme or a policy becomes part of organisational practices and policies), and individual-level indicators (monitoring of effectiveness of an intervention or programme 6 months or more after the most recent contact). | Seeing potential in Short message services |
| | | Possibilities of text message reminders |

RE-AIM, Reach, Effectiveness, Adoption, Implementation and Maintenance.

### Text message reminders support adherence

They described that before the intervention, their medication was often forgotten, which resulted in it being taken later than it should have been due to forgetfulness. Sometimes the adolescent realised that they had forgotten to take their medication but then waited until dinner to take it. The adolescents described that text message reminders were associated with taking their medicine and delays were avoided. Also, family members described situations when the adolescent's medication routines were interrupted, and where text message reminders played a big role.

> …Yes, it supports treatment adherence, because most of time we follow the television program but sometimes the power goes off, we missed the exact time to see medication time. Currently this text message comes 10 to 15 minutes before our medication time. (Father, aged 45)

### Feeling positive about the text messages reminder

The text message reminders contributed to positive attitudes and to decreased stigmatisation for adolescents, even when HIV/AIDS stigma is deeply rooted in the culture. Thus, the perception of the adolescents was that the text messages reminders played an important role in their overall well-being because sometimes they had low moods. The text message reminders were described as offering hope and supported the adolescents in achieving consistency in their medication adherence which helped them avoid, for example, fatigue.

> …The benefit of this message is it comes always and reminds me to take the drug on time. The other benefit is when I receive the message, I feel good. (Adolescent, aged 13)

### The code-delivered text message reminders minimise the risk of stigma

The adolescent felt safe regarding privacy in the text message reminders due to a code sent out to their telephone. The healthcare provider noticed a change in the views of both adolescents and family regarding increased trust and improved engagement with healthcare facilities during the intervention. Family expressed receiving coded text message reminders could enable adolescents to live like other HIV-free adolescents. However, the adolescents indicated that after their family's HIV status was made public, they were subjected to stigma, which was reflected in the interview. Some members of the families were open with their HIV disclosure while others were not.

> My status has already been made public, and everybody knows about my family as well … However, my husband lives in another area because of work, and knows his HIV status and makes things secret. He did not even take his medication when he come back home. (Mother, aged 45)

## Adoption

In this study, 11 health facilities with 153 participants received an intervention across 5 different geographic areas. According to healthcare providers, the rate of retention among adolescents has increased since the study began in the hospital, as described in the qualitative findings under a subtheme.

> …. By the way, in terms of retention, we have observed the best retention rate ever in this year since ART started in this hospital. Of course, our follow-up is there and there are other partners like CDC and ICAP. But this messaging has augmented the service we provide, and it increased our communication with them and helped them remind their medication irrespective of inconsistencies in messaging and medication time; the message delivery reminds them something targeted. (healthcare provider, aged 36)

### Supporting the text message reminders

The healthcare providers emphasised that text message reminders were an important support for adolescents in their medication intake to avoid poor adherence. In addition to this, families indicated that they had a mission to support the adolescents in avoiding ART treatment fatigue. Several adolescents expressed that they had received support from their family concerning medication intake during the intervention. Thus, the text message reminders were perceived as helpful, as they made participants feel happy and hopeful, even if they felt angered or were in a depressed mood that day.

> …The messages may help people pass good days, despite days that do not carry good fortune. It therefore helps people have good days and reminds them of medication times. One day, for example, being consumed with some ideas, I did not recall that I have medication to take; it is this text message that triggered me take my medicine. (Adolescent, aged 19)

### Implementation

There were 306 mobile phones provided to adolescents during assignments into intervention and control groups. The advisory committee was involved in the implementation of the intervention using standard operational procedure. During the study, 30 700 SMS text messages were sent to 153 study participants in the intervention group (figure 1).

Overall, intervention fidelity was high with (30 510/30 700) or 99.38% successfully delivered SMS text messages during the intervention.

The non-receiving participants in the intervention group were tracked and cross-checked using individual participant codes from the server, and then approached by each health facility to confirm whether they were to be followed up.

Changing phone numbers/SIM cards, phones being damaged, lost or shared with others were identified as

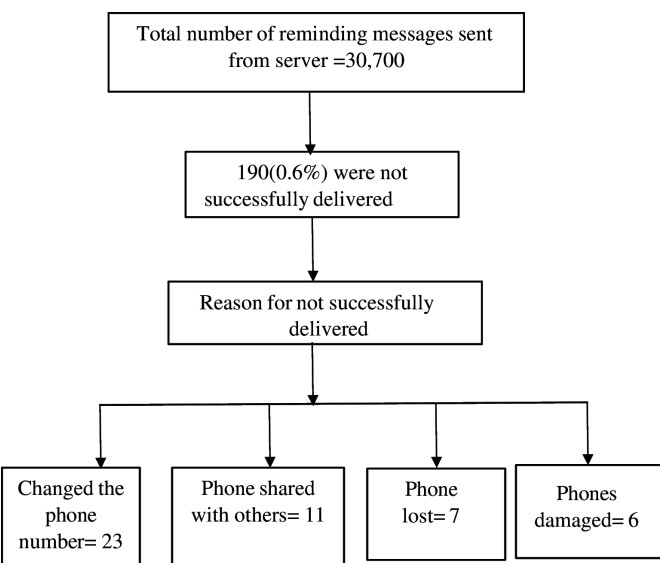

**Figure 1** Flow chart diagram. The flow chart shows the flow diagram of mobile phone text messaging reminder in the intervention group.

participant-related reasons for not receiving the messages and were registered in the logbooks, and corrective measures were used either to replace phones with new devices or repair/maintain the phones as soon as was feasible.

### Failing text message delivery influences medication intake

The adolescents identified several reasons for not receiving messages. In connection with the technical aspect, the adolescents highlighted problems such as not having a charged phone battery or having lost their phone. During the feasibility study period, there was fluctuation in messaging time, and the text message reminders arrived after their medication time. Another issue observed by both healthcare providers and family members was a missing SIM card. Furthermore, the continuity in the text message reminders was requested by the adolescents to continue the benefits.

> … the receiving time varies; sometimes I receive it 10 minutes before medication time, which is 3p.m. local time in the evening, and sometimes it arrives after I take the drug. (Adolescent, aged 13)

### Limitations in phone ownership

Sometimes adolescents were not permitted to use the telephone by themselves, due to restrictions from their family, as they thought that the adolescents were too young to use the telephone. Instead, the family used the telephone, which resulted in text message reminders never reaching the adolescent. The absence of SIM cards may be because adolescents did not receive a SIM card or because someone else in the family was using them. The family reported that they passed information from healthcare providers to the adolescents. Some of the families lacked information about the intervention and did not

know what to do when the telephone arrived. The healthcare providers emphasised that some families denied the adolescents use of the telephone, which could cause problems, and text message reminders were not received. Furthermore, the importance of informing families about the intervention was stressed, as several adolescents were restricted from using the telephone.

> …I think families should be communicated with regarding why their children are provided with mobile phones from the hospital, so as to avoid denial to access or adults using the phones for themselves. (Healthcare provider, aged 36)

### Technical limitations

One major issue stressed by adolescents, families and healthcare providers was that the electric power system went off quite often in their communities, which caused problems with receiving text message reminders if the battery in the telephone was not charged. The adolescents usually did not experience problems with delays in text message reminders, but they had observed that the text message reminders did not always arrive on time. The messages could come long before their scheduled medication time or sometimes very close to or after their scheduled medication time. Sometimes, the text message reminder was absent for entire days.

> …The message helped me a lot to remember my medication time. However, the skips in message delivery dates and untimely arrivals should be corrected so that it is more helpful. (Adolescent, aged 18)

### Maintenance

After initial configuration, a single server can send text messages to thousands of participants across a large geographic area. In this study, a server sent text messages to 153 participants across five different geographical areas. The technical challenges experienced in delivering the intervention were related to navigating the messaging server and feedback responses from the participants. The non-receiving participants in the intervention group were tracked and cross-checked using a four-digit individual participant code from the server and were then approached by each facility to confirm whether they were indeed in that group or lost. Despite technical barriers, and adverse conditions, and a belief in the future of text message reminders emerged, which was also expressed in the qualitative interviews.

### Seeing potential in SMS services

The benefits of SMS reminders were perceived as valuable, especially when adolescents missed appointments for medication refills and virus load tests. Before the intervention, there were difficulties getting in touch with the adolescents because they lived in rural areas and did not have a telephone. The healthcare providers thought about how to incorporate text message reminders into routine healthcare. A wish among the adolescents was expressed that more text message reminders could be helpful in raising awareness about their condition. Both the adolescents and the healthcare providers raised a need to continue the text message reminders.

> I think this must continue. The mobile provision has played significant roles in adherence and retention, as many of the adolescents do not have phones…because of this provision, adolescents can be reached directly with health messages and receive important lessons just from their phones. (Adolescent, aged 19)

### Possibilities with text message reminders

The healthcare provider described that possibilities to communicate with the adolescents had increased after the intervention, and the adolescents' attitudes towards healthcare facilities had become more positive. The adolescents found the improved digital communication to be an advantage because the healthcare provider no longer needed to call their family but could call them directly.

> Previously, healthcare providers from hospital reached me through neighbourhood phone numbers. She now uses this number to reach me directly through the phone provided by the hospital. This is a benefit for my privacy and for the information we discuss over the phone. (Adolescent, aged 18)

## DISCUSSION

Our process evaluation which was based on the RE-AIM framework[18] showed that intervention fidelity was high with only about 0.6% of the SMS messages not delivered as intended. Both providers and patients described the SMS reminders as being helpful. A few barriers were noted including glitches in telecommunication networks, electric power disruptions, issues with phone ownership at household levels. Our results highlight that SMS message reminders can be delivered with high coverage if proper processes are followed. The four-digit short code messaging system was a key enabler in ensuring confidentiality and security of the messages. Further scale-up of such interventions will depend on addressing broader systems issues including phone ownership, fixing network interruptions and reliable power sources. However, a study conducted in Philippines showed that intervention fidelity was low (77.9%) in the message group. The identified reason for low intervention fidelity was poor reliability of local telecommunication networks, and frequency of messages received.[28] The local context for cellular phone infrastructure and operational challenges, such as multiple users on a single cell phone, has an impact on text messaging interventions.[29] It is important to assess the setting before using the SMS intervention as a strategy to improve adherence. There is emerging evidence that mobile phones can play an important role

in healthcare delivery, especially in resource-limited settings.[30] SMS text messaging is a particularly useful application that can be used to collect or share information and to enhance communication between healthcare personnel and patients in a low-cost manner.[31] Treatment support was highlighted as an important factor for adolescents to avoid poor adherence. Text messages could be a facilitator for adolescents to remain hopeful and avoid unwanted side effects given that previous research[32 33] has shown that a lack of support systems was perceived to negatively affect adherence among young people. Our findings provide additional evidence in support of research to identify effective interventions to improve adherence and retention in care for adolescents with HIV and enhance their engagement in these services. Thus, findings of this process evaluation can guide further research towards interventions concerning readiness among healthcare facilities because settings can impact implementation. Giving importance to the voice of adolescents living with HIV can offer hope to many more in the same situation. This study thereby has a potential clinical impact and contributes to UN Sustainability Goal.[34]

### Methodological concerns/limitations

There are both strengths and limitations identified in this process evaluation. First, the intervention has several strengths, and a key strength is the use of a four-digit short code for text messages that enables rapid, high-volume outbound messaging, is easy to read and remember and assists text recipients in identifying the sender based on their preferences for message delivery time, message content and messaging language. The use of the unique code can contribute to maintaining security and confidentiality among participants.

Second, the messaging server is adaptable and user-friendly, as the researcher or local IT provider can add and remove new data fields to navigate dropped or missed individuals and update SMS text message content to avoid message-related fatigue.

In addition, the process evaluation also has several strengths that were identified. The first strength is the use of mixed methods, where different forms of data—both quantitative and qualitative methods—are combined to capture the multidimensionality of the intervention. The data supplement one another and aid in obtaining a comprehensive picture of the study results, which can contribute to a deeper understanding of the participants' experiences.[35] If the participants are given the opportunity to highlight their own individual priorities, it might be possible for us to improve our understanding of the impact of the intervention.

Finally, three different perspectives (adolescents, families and healthcare providers) were included in the qualitative interviews, which allowed different voices to be heard. However, the limited number of included participants from adolescents' families and healthcare providers can limit the study's transferability.[36] The way research results are presented can be influenced by

their nature and outcome, leading to what is known as reporting bias. To minimise outcome reporting bias, the study protocol was preregistered in the pan-African clinical trials registry. A composite adherence measurement was used, and Consolidated Standards of Reporting Trials recommendations were followed.[37] The fact that adherence data were self-reported by healthcare professionals within the same facilities through interviews may have a significant impact on the study's validity and make it prone to social desirability bias. However, in this study, the primary author conducted the interviews in a separate, private room. Furthermore, providing cell phones to adolescents during recruitment may reduce the likelihood of interview refusal in the intervention phase, causing acquiescence bias.

Trustworthiness is the most commonly used criterion for evaluating qualitative content analysis, and it is often expressed using terms such as credibility, conformability, dependability and transferability.[36] To ensure the credibility of the study, an appropriate data collection method needs to be used.[38] This study used separate interview questions for adolescents, family and professional interviews to explore their experiences and issues with mobile text messaging interventions. In this study, the five-dimension components of the RE-AIM framework were used to collect information about the intervention setting, implementation, personnel and circumstances and findings by considering the individual level and external context challenges.[18] To ensure the dependability of the study, an interview guide adapted to the participants, and audio recorded was used. For the confirmability of this study, the first author transcribed the audio data to verbatim text and translated from Amharic to English by considering both manifest and latent content. Then, all authors involved in reading the transcripts, extracted meaning units, codes and data analysis.

Finally, the author's previous qualitative research experiences and knowledge, as well as a bracketing technique were used to increase the trustworthiness of their findings. Accordingly, the primary author maintained a positive relationship with participants during intervention follow-ups and established independence from caregiving facilities to minimise social desirability bias and boss views. To reduce bias and subjective interpretation, interviews were audio recorded, transcribed and thoroughly evaluated, with participant quotes used to support the authors' findings.

## CONCLUSION

The mobile phone text messaging intervention to support ART adherence and retention in care was well received by participants, and overall intervention fidelity was high. However, the intervention feasibility was dependent on the reliability of local telecommunication networks, local electric power and monitoring of adolescents' families, all of which had a significant impact on the intervention usability, fidelity and doses received. The participants

experienced the text messaging function as helpful. Therefore, the development of different aspects, such as addressing technical problems, would be desirable to further improve impact. Further research needs to be directed towards readiness within healthcare facilities to capture an increased understanding of barriers and facilitators in implementation.

**Acknowledgements** The authors sincerely thank the participants for sharing their experiences, and all staff working in the ART clinics for facilitating the study.

**Contributors** AT, DJ and IH conceptualised and developed the study design. AT collected the data and transcribed verbatim and translated from Amharic to English. DJ and IH supervised data collection and study implementation. All authors read the transcripts, extracted meaning units, codes and data analysis. Subthemes were created from the code frame by the first and last authors. AT, HÅP, DJ, and IH are guarantor. AT and HÅP drafted the manuscript, and all authors contributed to critical revision and approved the final manuscript.

**Funding** The project was externally funded by FORTE: Swedish Research Council for Health, Working Life and Welfare Life (FORTE) (https://forte.se/en/) program support 2018-01399 and the Swedish Research Council(https://www.vr.se/english.html) programme support VR 2016-05706. The study protocol has undergone a peer review by the funding body.

**Disclaimer** The funders had no role in study design, writing of the study protocol, the decision to publish the study protocol, or preparation of the manuscript.

**Competing interests** None declared.

**Patient and public involvement** Patients and/or the public were not involved in the design, or conduct, or reporting, or dissemination plans of this research.

**Patient consent for publication** Not applicable.

**Ethics approval** This study involves human participants and ethical approval was obtained from the Swedish Regional Ethical Review Board (Dnr 2019-03433), National Research Ethics Review Committee in Ethiopia (MoSHE/RD/142/2869/20) and the Institutional Research Ethics Review Board in Arba Minch University (IRB-113/11). Participants gave informed consent to participate in the study before taking part.

**Provenance and peer review** Not commissioned; externally peer reviewed.

**Data availability statement** Data are available on reasonable request. Extra data can be accessed via the Dryad data repository at https://doi.org/10.5061/dryad.n8pk0p31s.

**ORCID iD**
Abayneh Tunje http://orcid.org/0000-0002-0957-0216

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
