## [Reviewer comments · BMJ Open]

ARTICLE DETAILS

TITLE (PROVISIONAL)	Intervention fidelity and factors affecting the process of implementing a mobile phone text messaging intervention among adolescents living with HIV: a convergent mixed methods study in southern Ethiopia
AUTHORS	Tunje, Abayneh; Persson, Helene; Jerene, Degu; Hallstrom, Inger

VERSION 1 – REVIEW

REVIEWER	Rashmi Rodrigues St Johns Medical College, Community Health
REVIEW RETURNED	19-Sep-2023

GENERAL COMMENTS	Dear Authors, First of all, I commend the authors on the interesting and timely research undertaken and the use of the RE-AIM framework to evaluate the implementation of the intervention i.e., text messages to support adherence to antiretroviral therapy in adolescents in Ethiopia. However, the mixed methods approach used is not very clear as there is no quantitative analysis described and the quantitative results are very brief and therefore whether this is actually a mixed methods paper needs to be thought through. Title: ...process of a mobile phone text messaging intervention... What process does the title refer to?- process of implementation? Abstract: To be modified based on changes in the manuscript. Introduction: Page 6: Line 3-6. The same intervention may have different effects in different contexts, due to weaknesses in its design or improper implementation- while there is a reference for this statement, it may not be entirely true as the success of interventions is multifactorial and may depend on factors such as sociodemography, culture etc. in addition to its design and implementation may be- hence, kindly rephrase. Methodology: Description of the intervention- an example of a few text messages used would be useful. Also, what language was the intervention delivered in? were multiple languages used? What was the literacy level of the participants as well as their familiarity with mobile phones (owned or shared) prior to participating in the study?
---

	There is a 4-digit code that is much described in the results. However, the description in the methods is unclear. ... an automated message pushing software was used to push tailored daily text messages through a 4-digit code... Study setting and participants: A table with the description of the participants in the qualitative part of the study is necessary in terms of sociodemography and their role in the main study. What were the inclusion and exclusion criteria? Were these adolescents living at home or in hostels? Sample and data collection: What was the basis for purposive sampling? Please describe the process of qualitative data collection as these seem to be rather lost in the process of focussing on the RE-AIM framework i.e., who conducted the interviews? where were the interviews conducted- what was the environment especially related to privacy and confidentiality? how long did the interviews last? Which language were the interviews conducted in? Who conducted the interviews? (though not necessarily in this order) Privacy and confidentiality of data management? Data analysis: Page 9, lines 19 to 36: This should ideally be described along with the description of the intervention or under the subheading intervention fidelity.- what you have described here- the capturing or delivered and undelivered messages represents how data was collected regarding intervention fidelity This should ideally describe both the quantitative and qualitative data analysis methods. There does not seem to be any quantitative data analysis described. Was any software used for qualitative data analysis? Which authors were involved in the analysis- please indicate. Ethical Concerns/ statement: There is none described, hence please add a statement on IEC clearance, consent, and confidentiality. Results: Reach: Needs a description of how the participants enrolled differed from those not enrolled (excluded or refused consent), to understand the actual reach of the intervention. Effectiveness: Please include a brief statement on the effectiveness of the intervention used here, prior to presentation of qualitative results. Page 13, line 18: To clarify- did the participants feel that the intervention 'increased adherence' or did they feel that the intervention 'supported adherence' to medications?- from quote below- the inference is it 'supports adherence' Line 49: Feeling comfortable receiving text message reminders:- the text under this theme does not really describe the theme- hence kindly consider revising this theme. Page 14, Line 18/19: The code-delivered text messages reminders minimises the risk of stigma: it is a little unclear how disclosure of HIV status of the families is associated with the theme. Also, the statement- However, the adolescents experienced that far from everyone talked openly about their HIV disclosure, which was also reflected in the interviews, needs to be simplified or rephrased. Adoption: The theme- treatment support does not reflect adoption.
--	---

	Implementation: Page 14: Lines 20-53: Overall, intervention fidelity was high with (30,510/30,700) or 99.38% successfully received SMS text messages during the intervention. It is not clear how intervention receipt was assessed in your methods section. You do mention that intervention delivery was tracked would it be the same as intervention receipt? Could there be reasons why an intervention that was delivered was not received? Page 16, line 33-38: The adolescents usually did not experience problems with delays in text messages reminders, but they had observed that the text messages reminders did not always arrive on time- the statements seem contradictory here. Page 18, lines 12-17: Despite technical barriers, and adverse conditions, and a belief in the future of text messages reminders emerged, which was also expressed in the qualitative interviews – were there any other kind of assessments of acceptability of intervention conducted- this especially as you say- ‘which was also expressed in the qualitative interviews.’ If yes, please describe all assessment procedures in the methods section. The RE-AIM framework also include an assessment of costs for implementation- which I do not see here. Discussion: Please include a section on methodological concerns/ limitations –in terms of methods and results. Also please include a section on the role of each author in the qualitative part of the study- which is generally included in the methods section. Further, trustworthiness and reflexivity- especially the authors role in data collection, analysis and interpretation of findings needs to be described. It appears that the interviews are based on participants who were retained in the study – and therefore presents only positive experiences. Was there no participant who had problems with the intervention in terms of stigma/ confidentiality or other problems other than – not receipt of messages. Please check the language for minor edits. Best wishes!
--	---

REVIEWER	Deborah Rinehart Denver Health
REVIEW RETURNED	27-Oct-2023

GENERAL COMMENTS	Using mixed methods this study attempted to provide more detail on the implementation and outcomes of a text messaging intervention. However, the manuscript was hard to follow, and the two methods were not well integrated. Introduction On page 5 one of the aims is to explain the outcomes of the trail. I do not think this was covered in the manuscript and if it was it would be helpful to clearly call this out in the results section. Methods More detail is needed to better understand the methods for this study. For example, it is unclear how adolescents were recruited into and the data that was captured in the quantitative piece. A 6-month follow-up is alluded to in the results but not presented in the methods. The qualitative evaluation is adequately explained. The analysis section regarding the SMS messages has information that could be better presented in the methods versus the analysis. The analysis should contain the variables (outcomes) and response
--

	options and the analyses that were used – e.g., descriptive analyses or chi-square/t-tests. The results are structured following the RE-AIM framework perhaps the analyses section could follow this same format so it is clear what variables are being using to measure what. Results Reach – Eligibility criteria was never mentioned prior to this section. Also the reach section seems to assess study recruitment for the RCT and not reach of the intervention. Effectiveness –As the methods section did not clearly state the outcome variables for effectiveness it was hard to understand the results. On page 11, line 37 this is the study retention number not an effectiveness outcome. Additionally, page 11 line 37 is the first time there is mention of a 6-month follow-up. Adoption – The second sentence is confusing. Consider including a provider quote here related to adoption at the healthcare “setting” level. Implementation – clarify why there were 306 phones provided to 153 intervention group participants. Page 15 line 6 indicates not all participants were followed up, this was confusing and it was not clear what follow up entails. Clarify if the data on changing phones (page 15 starting on line 11) was from the qualitative interviews or from a different follow up interview. As this paragraph preceded the qualitative themes it was unclear where these data were from. Discussion The 4-digit code is discussed in the discussion and highlighted as a strength, but it was not explained in the intervention description. Provide more detail on this and how it worked for the end user. Tables Perhaps add quantitative results to Table 1 as this might help to better integrate the two methods.
--	---

VERSION 1 – AUTHOR RESPONSE

However, the mixed methods approach used is not very clear as there is no quantitative analysis described and the quantitative results are very brief and therefore whether this is actually a mixed methods paper needs to be thought through.

Response: We thank the editor for commending our work. We also appreciate the editor’s comment on the quantitative data. We added more details about the quantitative data in the methods and results sections (page 10, lines 218-219 for analysis), and page 12, line 250 in table 1 and page 14, lines 263-266 for quantitative results respectively.

Title: ...process of a mobile phone text messaging intervention... What process does the title refer to?- process of implementation?

Response: we appreciate this important comment on the title. The manuscript was about the process of implementation. We therefore added “the process of implementing” to the title and amended title reads as “Intervention fidelity and factors affecting the process of implementing a mobile phone text messaging intervention among adolescents living with HIV: a convergent mixed methods study in southern Ethiopia.”

Abstract: To be modified based on changes in the manuscript.

Response: Abstract modified based on changes in the manuscript.

Introduction:

Page 6: Line 3-6. The same intervention may have different effects in different contexts, due to weaknesses in its design or improper implementation- while there is a reference for this statement, it may not be entirely true as the success of interventions is multifactorial and may depend on factors

such as sociodemographic, culture etc. in addition to its design and implementation may be- hence, kindly rephrase.

Response: We thank the reviewer for this important comment. We corrected and described in page 5, lines 94-95 as follows; "In addition to design and implementation, the success of interventions is diverse and may be influenced by factors such as sociodemographic characteristics and culture."

Methodology:

Description of the intervention- an example of a few text messages used would be useful.

Response: We thank the reviewer for this helpful suggestion.

We added the following description in page 8, lines 172-174.

The type of message was short and mainly focused on advice about the benefits of taking on time, the risk/consequences of missing/not taking on time, encouragement, and without mentioning anything about HIV or the name of the drug. For example, "Those who use it properly know its benefits", "I'm taking it properly and on time". The Amharic equivalents of these messages were "በትክክል የሚጠቀሙት ጥቅሞችን ያውቃሉ" and በትክክል እና በሰዓት እውስዳለሁ respectively.

Also, what language was the intervention delivered in? Were multiple languages used?

Response: Adolescents were asked to choose the language in which they wanted to receive the reminder text message and the messaging time to remind them of the medication time. All of them chose Amharic language, which is one of the national languages and the language the intervention was delivered in.

What was the literacy level of the participants as well as their familiarity with mobile phones (owned or shared) prior to participating in the study?

Response: One of our requirements for the main study was being able to read text. Those who were unable to read were excluded from the study (page 6, lines 136-137). Regarding mobile phone ownership and use, we provided mobile phones to all study participants. How to use the mobile device, including saving, and deleting messages was demonstrated to all (page 7, lines 142-143).

There is a 4-digit code that is much described in the results. However, the description in the methods is unclear. ... an automated message pushing software was used to push tailored daily text messages through a 4-digit code...

Response: We thank the reviewer for this valid comment. We added more details about the 4-digit code in the methods section, on page 7, lines 160-161 and page 8, lines 162-164, as follows: "After the intervention's development, text messages were uploaded to the server, and an automated message-pushing software was used to send tailored daily text messages using a four-digit code from Ethiopia Telecom. This code helped adolescents to distinguish intervention messages from other messages."

Study setting and participants: A table with the description of the participants in the qualitative part of the study is necessary in terms of sociodemography and their role in the main study.

Response: This comment is well noted. On page 11, line 250 in table 1, we have included the number of males/females, language, and the geographic areas they represent in the manuscript. Due to confidentiality reasons further sociodemographic variables are not presented.

What were the inclusion and exclusion criteria?

Response: We added the inclusion and exclusion criteria in the methods section, on page 6, lines 133-138 and page 7, lines 139-143 as follows: "We used the following technique to identify eligible adolescents: first, we identified the number of adolescents of eligible age in each facility. Then, using their address, we called each adolescent and their parents or caregivers directly via phone numbers, care providers, or adherence supports. Finally, data collectors approached adolescents and posed literacy questions about their ability to read text by displaying sample texts. Then, eligible participants in the study were asked to give consent and assent. Thus, adolescents aged 10 to 19 years, diagnosed with HIV, currently participating in ART care, and planning to stay in the designated facility for at least six months following study enrolment were eligible. Participants who were above the age limit, did not disclose their HIV status, were unable to read text, or had a disability were all excluded.

Finally, the data collector showed each adolescent how to use their mobile device, including text saving and deletion.”

Were these adolescents living at home or in hostels?

Response: We thank the reviewer for this. As described in Table 1, page 12, only 10 (6.5%) of the participants lost one or more parents. Of these, only two adolescents were under institutional care. Sample and data collection: What was the basis for purposive sampling?

Response: We thank the reviewer for this important question. Initially, in the main trial computer generated simple random sampling was used. However, in the process evaluation purposive sampling was used meaning that participants from different geographic areas, gender, ages. Also, different participants involved in the intervention included i.e. adolescents, parents/caregivers, and health care providers. (page 9, lines 191-193)

Please describe the process of qualitative data collection as these seem to be rather lost in the process of focusing on the RE-AIM framework i.e., who conducted the interviews?

Response: The interviews were conducted by the first author and one research assistant and is described in page 9 line 196.

Where were the interviews conducted- what was the environment especially related to privacy and confidentiality?

Response: The interviews were conducted in a quiet and confidential area at the respective health facilities and is described in page 9 lines 195-196

How long did the interviews last?

Response: Each interview lasted on average 30 minutes and is described in page 9 line 197.

Which language were the interviews conducted in?

Response: The interviews were conducted in all respondent languages, such as Amharic, Wolaytigna, Gamogna, Gofigna, Arigna, and Konsegna and is described in page 9 line 195.

Who conducted the interviews? (though not necessarily in this order) Privacy and confidentiality of data management?

Response: The interviews were conducted in a quiet and confidential area at the respective health facilities by the first author and one research assistant. Please see answer above.

Data analysis:

Page 9, lines 19 to 36: This should ideally be described along with the description of the intervention or under the subheading intervention fidelity. - what you have described here- the capturing or delivered and undelivered messages represents how data was collected regarding intervention fidelity This should ideally describe both the quantitative and qualitative data analysis methods.

There does not seem to be any quantitative data analysis described.

Response: We thank the reviewer for this important concern. As we described in page 10 lines 218-220, The descriptive statistical analysis was used to summarize data in frequency, percentages, mean and standard deviation. The quantitative effectiveness data is analyzed separately and not included in the process evaluation.

Was any software used for qualitative data analysis?

Response: The first author developed a coding frame for the analysis by using NVivo software including meaning units and codes, which were discussed by all authors and is mentioned in page 10, line 228.

Which authors were involved in the analysis- please indicate.....

Response: As described in pages 10 lines 224-228 and page 11 lines 229-234 “Interviews were analysed by using the principles for content analysis described by Graneheim and Lundman. First, the transcripts were read several times to gain a sense of the whole by the first and the last author. Thereafter, meaning units were identified and condensed to reduce the text while maintaining the core meaning. The first author developed a coding frame for the analysis by using NVivo software, which were discussed by all authors. Condensed meaning units were then labelled with a code, which were kept close to the text on a manifest level. Next, sub-themes were created from the code frame by the

first and last authors. The sub-themes were then abstracted into themes. Finally, the underlying meaning—the latent content—was described into a main theme. Thereafter one of the authors (HÅP) read the transcripts, extracted meaning units, codes, and themes. The content was discussed and reflected upon together with all authors and lasted until an agreement was reached.”

Ethical Concerns/ statement: There is none described, hence please add a statement on IEC clearance, consent, and confidentiality.

Response: This comment is well noted. We included consent and confidentiality statements on page 25, lines 511-589 described as follows: “The study was conducted according to the guidelines in the Declaration of Helsinki (37) and approved by the Swedish Regional Ethical Review Board (Dnr 2019-03433), National Research Ethics Review Committee in Ethiopia (MoSHE/RD/142/2869/20), and the Institutional Research Ethics Review Board in Arba Minch University (IRB-113/11). The participants received both oral and written information about the purpose of the study, about the confidential treatment of the data and the voluntary nature of participation. Informed consent was provided from all participants over the age of 18 before the study began. Similarly, parental consent and assent were obtained from all adolescents under the age of 18 before the start of the trial.”

Results:

Reach:

Needs a description of how the participants enrolled differed from those not enrolled (excluded or refused consent), to understand the actual reach of the intervention.

Response: We thank the reviewer for this comment. on page 14, lines 259-262 described as follows: “The remaining 129 participants were excluded for the following reasons: 27 (20.9%) were unable to stay in the study area for the duration of the study, 39 (30.2%) were unable to read, 24 (18.6%) did not know their status, 21 (16.3%) could not be reached via phone, 13 (10.1%) refused to participate, and 5 (3.9%) had an impairment.

Effectiveness:

Please include a brief statement on the effectiveness of the intervention used here, prior to presentation of qualitative results.

Response: Thank you for your suggestion to brief more about effectiveness in page 10 lines 219-220, we mention about the effectiveness as “the quantitative effectiveness data is analyzed separately and not included in the process evaluation”

Page 13, line 18: To clarify- did the participants feel that the intervention ‘increased adherence’ or did they feel that the intervention ‘supported adherence’ to medications? - from quote below- the inference is it ‘supports adherence’

Response: We thank the reviewer for this constructive question.

The respondents felt it “supports adherence” and we corrected the statements of effectiveness based on your suggestion on page 15, line 280.

Line 49: Feeling comfortable receiving text message reminders: - the text under this theme does not really describe the theme- hence kindly consider revising this theme.

Response: We agree with the reviewer’s suggestion and changed the theme in page 15, line 292 as follows;

“Feeling comfortable receiving text message reminders”to “Feeling positive about the text messages reminder”

Page 14, Line 18/19: The code-delivered text messages reminders minimises the risk of stigma: it is a little unclear how disclosure of HIV status of the families is associated with the theme.

Also, the statement- However, the adolescents experienced that far from everyone talked openly about their HIV disclosure, which was also reflected in the interviews, needs to be simplified or rephrased.

Response: We amended the sentence as per the reviewer’s comment and is described in page 16, lines 307-310 as “However, the adolescents indicated that after their family’s HIV status was made public, they were subjected to stigma, which was reflected in the interviews.”

Adoption:

The theme- treatment support does not reflect adoption.

Response: Thank you for your constructive comment. We have changed the theme to Supporting the text messages reminders (page 17, line 326)

Implementation:

Page 14: Lines 20-53: Overall, intervention fidelity was high with (30,510/30,700) or 99.38% successfully received SMS text messages during the intervention.

It is not clear how intervention receipt was assessed in your methods section. You do mention that intervention delivery was tracked, would it be the same as intervention receipt?

Response: We thank the reviewer for this question. Individual participant codes from the server were used to track and cross-check sent or unsuccessful messages for intervention fidelity. In addition, during the follow-up interviews, adolescents were asked about receiving reminder text messages on their phones and we incorporated this statement in the data collection section under method section, on page 9, lines 189-190 described as "During the follow-up data collection phase, adolescents were also asked if they received text messages on a regular schedule."

Could there be reasons why an intervention that was delivered was not received?

Response: As we mentioned in the section on non-receiving participants (on page 18 lines 365-366), one of the identified reasons for delivered message not being received by adolescents was that some of the adolescents was that other people used their phones. This happened in situations where their family or caregiver was not biological family and if the adolescents were in early adolescence (age 10-14 years). This question is addressed in the "limitations in phone ownership" theme.

Page 16, line 33-38: The adolescents usually did not experience problems with delays in text messages reminders, but they had observed that the text messages reminders did not always arrive on time- the statements seem contradictory here.

Response: During the feasibility study, text message reminders did not always arrive on time, and there was a time zone shift while sending text messages. After fixing this technical glitch based on the feasibility data, no more problems occurred when the main trial began. In this manuscript, we included data from the feasibility period. To make this clearer, we revised the text on page 18, lines 356-357 as follows: " During the feasibility study period, there was fluctuation in messaging time, and the text message reminders arrived after their medication time."

Page 18, lines 12-17: Despite technical barriers, and adverse conditions, and a belief in the future of text messages reminders emerged, which was also expressed in the qualitative interviews – were there any other kind of assessments of acceptability of intervention conducted- this especially as you say- 'which was also expressed in the qualitative interviews.'

Response: There were no other assessments of the acceptability of the intervention. However, qualitative interviews with adolescents were done regarding the benefits of text message reminders. If yes, please describe all assessment procedures in the methods section.

The RE-AIM framework also includes an assessment of costs for implementation- which I do not see here.

Response: Thank you very much for your concern. We did not include cost related information on this manuscript.

Discussion:

Please include a section on methodological concerns/ limitations –in terms of methods and results.

Response: We agree with the reviewer and included a section on methodological limitations on page 23. lines 469-471 as follows:" However, limited numbers of included participants from adolescents' families and healthcare providers can limit the results transferability."

Also please include a section on the role of each author in the qualitative part of the study- which is generally included in the methods section. Further, trustworthiness and reflexivity- especially the author's role in data collection, analysis and interpretation of findings needs to be described.

Response: We included this in page 24, lines 492-49 as follows:

AT, DJ, and IKH conceptualized and developed the study design. AT collected the data and transcribed verbatim and translated from Amharic to English. DJ and IKH supervised data collection and study implementation. All authors read the transcripts, extracted meaning units, codes, and data analysis. Sub-themes were created from the code frame by the first and last authors. AT and HÅP drafted the manuscript, and all authors contributed to critical revision and approved the final manuscript.

It appears that the interviews are based on participants who were retained in the study – and therefore present only positive experiences. Was there no participant who had problems with the intervention in terms of stigma/ confidentiality or other problems other than – not receipt of messages.

Response: Yes, that is correct. No participant reported about stigma or concerns about confidentiality from each facility and during follow data collection after enrolling into the study.

Please check the language for minor edits.

Response: language is checked for minor edits.

Best wishes!

Reviewer: 2

Dr. Deborah Rinehart, Denver Health

Comments to the Author:

Using mixed methods this study attempted to provide more detail on the implementation and outcomes of a text messaging intervention. However, the manuscript was hard to follow, and the two methods were not well integrated.

Introduction

On page 5 one of the aims is to explain the outcomes of the trial. I do not think this was covered in the manuscript and if it was it would be helpful to clearly call this out in the results section.

Response: We accept the reviewer's valuable comment.

The statement was deleted from the document because it was considered unnecessary for the position. "This information can be used to explain the outcomes of the trial and can guide the development and evaluation of similar interventions."

Methods

More detail is needed to better understand the methods for this study. For example, it is unclear how adolescents were recruited and the data that was captured in the quantitative piece. A 6-month follow-up is alluded to in the results but not presented in the methods. The qualitative evaluation is adequately explained.

Response: We had similar comments from reviewer 1 and thus added more details about the participant recruitment on page 6, lines 133-138 and page 7, lines 139-143: modified your comment about adolescents' recruitment and included it to the method section as follows "We used the following technique to identify eligible adolescents: first, we identified the number of adolescents of eligible age in each facility. Then, using their address, we called each adolescent and their parents or caregivers directly via phone numbers, care providers, or adherence supports. Then, data collectors approached adolescents and posed literacy questions about their ability to read text by displaying sample texts. Then, participants in the study were asked to give consent and assent. Thus, adolescents aged 10 to 19 years, diagnosed with HIV, currently participating in ART care, and planning to stay in the designated facility for at least six months following study enrolment were eligible. Participants who were above the age limit, did not disclose their HIV status, were unable to read text, or had a disability were all excluded. Finally, the data collector showed each eligible adolescent how to use their mobile device, including text saving and deletion."

We also included more details about quantitative data (page 9, lines 186-191) described as follows "The quantitative data was collected from the server, activity logs, an adverse event monitoring form,

and an interview with adolescents. The researcher collects data on intervention fidelity by tracking the number of text messages sent, received, and failed from the server daily. Furthermore, each facility representative documented any adverse event related to the trial, filled the form, and sent it to researchers every week. Furthermore, during the follow-up data collection phase, adolescents asked if they received text messages on a regular schedule.”

The analysis section regarding the SMS messages has information that could be better presented in the methods versus the analysis. The analysis should contain the variables (outcomes) and response options and the analyses that were used – e.g., descriptive analyses or chi-square/t-tests.

Response: We thank the reviewer for this comment. We added further description in the methods section (page 10 lines 218-220): “The descriptive statistical analysis was used to summarize data in frequency, percentages, mean and standard deviation. The quantitative effectiveness data is analyzed separately and not included in the process evaluation.”

The results are structured following the RE-AIM framework. Perhaps the analyses section could follow this same format, so it is clear what variables are being used to measure what.

Response: thank you very much for your important points and we incorporated in analysis section in page 10 lines 220-223 as follow “The RE-AIM outcomes framework components of Reach, Effectiveness, Adoption, Implementation, and Maintenance were used to present mixed method findings about the delivery of a reminder message, success, and intervention barriers for the intervention implementation process”.

Results

Reach – Eligibility criteria was never mentioned prior to this section. Also the reach section seems to assess study recruitment for the RCT and not reach of the intervention.

Response: We thank the reviewer for this comment and is described in page 6, lines 133-138 and page 7, lines 139-143 as follows: We used the following technique to identify eligible adolescents: first, we identified the number of adolescents of eligible age in each facility. Then, using their address, we called each adolescent and their parents or caregivers directly via phone numbers, care providers, or adherence supports. Then, data collectors approached adolescents and posed literacy questions about their ability to read text by displaying sample texts. Then, participants in the study were asked to give consent and assent. Thus, adolescents aged 10 to 19 years, diagnosed with HIV, currently participating in ART care, and planning to stay in the designated facility for at least six months following study enrolment were eligible. Participants who were above the age limit, did not disclose their HIV status, were unable to read text, or had a disability were all excluded. Finally, the data collector showed each eligible adolescent how to use their mobile device, including text saving and deletion. However, for the intervention arm, the reach dimension addresses 153 adolescents.

Effectiveness –As the methods section did not clearly state the outcome variables for effectiveness it was hard to understand the results. On page 11, line 37 this is the study retention number not an effectiveness outcome. Additionally, page 11 line 37 is the first time there is mention of a 6-month follow-up.

Response: The effectiveness result is under preparation to be reported separately.

Adoption – The second sentence is confusing. Consider including a provider quote here related to adoption at the healthcare “setting” level.

Response: This comment is well noted. We incorporated it in page 16 lines 320-325 as follow. According to healthcare providers, the rate of retention among adolescents has increased since the study began in the hospital, as described in the qualitative findings under a sub-theme. “...By the way, in terms of retention, we have observed the best retention rate ever in this year since ART started in this hospital. Of course, our follow-up is there and there are other partners like CDC and ICAP. But this messaging has augmented the service we provide, and it increased our communication with them and helped them remind their medication irrespective of inconsistencies in messaging and medication time; the message delivery reminds them something targeted”.

Implementation – clarify why there were 306 phones provided to 153 intervention group participants.
 Response: We apologize for the typo. As we described in page 17, lines 340-341, we provided 153 phones to the 153 participants in the intervention group. 306 phones were for the control group inclusive.

Page 15 line 6 indicates not all participants were followed up, this was confusing, and it was not clear what follow up entails. Clarify if the data on changing phones (page 15 starting on line 11) was from the qualitative interviews or from a different follow up interview. As this paragraph preceded the qualitative themes it was unclear where these data were from.

Response: Thank you very much for your important concern. It is from the qualitative interview.

Discussion

The 4-digit code is discussed in the discussion and highlighted as a strength, but it was not explained in the intervention description. Provide more detail on this and how it worked for the end user.

Response: We agree with the reviewer and described this in the method section on page 7, lines 160-161 and page 8, lines 162-164 as follow, “After the intervention’s development, text messages were uploaded to the server, and an automated message-pushing software was used to send tailored daily text messages using a four-digit code from Ethiopia Telecom. This code helps adolescents in avoiding messaging source-related stigma and distinguishing intervention messages from other messages.”

Tables

Perhaps add quantitative results to Table 1 as this might help to better integrate the two methods.

Response: Agreed and added quantitative results in table 1 in page 12, line 250.

VERSION 2 – REVIEW

REVIEWER	Rashmi Rodrigues St Johns Medical College, Community Health
REVIEW RETURNED	10-Jan-2024

GENERAL COMMENTS	Dear Authors, Thank you for the detailed response. The efforts to address the reviewer's comments are well appreciated. I have a few minor comments for your consideration:  1. The methodological issues/ limitations are again not discussed in terms of the methods used in the paper- kindly discuss in terms of biases- reporting bias, acquiescence, and social desirability that may have affected the responses. 2. Trustworthiness in a qualitative study is generally described as credibility, transferability, dependability, and confirmability- and has not been addressed adequately. 4. Reflexivity in qualitative research does not only mean who did what but also how the researcher's prior experience with the topic (not necessarily only as a researcher but also as an individual) has influenced the interpretation of qualitative data. Best wishes!
---

VERSION 2 – AUTHOR RESPONSE

Reviewer: 1

Comments to the Author:

1. The methodological issues/ limitations are again not discussed in terms of the methods used in the paper- kindly discuss in terms of biases- reporting bias, acquiescence, and social desirability that may have affected the responses.

Response: The authors would like to thank the reviewer for pointing this out. We addressed the reviewer's valuable comment, and added more details about the reporting bias, acquiescence, and social desirability bias were included in page 21 line 432-440 as "The way research results are presented can be influenced by their nature and outcome, leading to what is known as reporting bias. To minimize outcome reporting bias, the study protocol was pre-registered in the pan-African clinical trials registry. A composite adherence measurement was used, and CONSORT recommendations were followed (37). The fact that adherence data was self-reported by healthcare professionals within the same facilities through interviews may have a significant impact on the study's validity and make it prone to social desirability bias. However, in this study, the primary author conducted the interviews in a separate, private room. Furthermore, providing cell phones to adolescents during recruitment may reduce the likelihood of interview refusal in the intervention phase, causing acquiescence bias."

2. Trustworthiness in a qualitative study is generally described as credibility, transferability, dependability, and confirmability- and has not been addressed adequately

Response: We agree with the reviewer's valuable comment.

We added more details about the reporting bias, acquiescence, and social desirability bias were included in page 21 and 22 line 441-452 as "Trustworthiness is the most commonly used criterion for evaluating qualitative content analysis, and it is often expressed using terms such as credibility, conformability, dependability, and transferability (36). To ensure the credibility of the study, an appropriate data collection method needs to be used (38). This study used separate interview questions for adolescents, family, and professional interviews to explore their experiences and issues with mobile text messaging interventions. In this study, the five-dimension components of the RE-AIM framework were used to collect information about the intervention setting, implementation, personnel and circumstances, and findings by considering individual level and external context challenges (18). To ensure the dependability of the study, an interview guide adapted to the participants, and audio-recorded was used. For the confirmability of this study, the first author transcribed the audio data to verbatim text and translated from Amharic to English by considering both manifest and latent content. Then, all authors involved in reading the transcripts, extracted meaning units, codes, and data analysis."

3. Reflexivity in qualitative research does not only mean who did what but also how the researcher's prior experience with the topic (not necessarily only as a researcher but also as an individual) has influenced the interpretation of qualitative data.

Response: We are grateful for the reviewer's valuable comment. This was partially addressed in our response to comment #1, but we added more details about the reflexivity of authors were included in page 22 line 453-458 as "Finally, the authors previous qualitative research experiences and knowledge, as well as a bracketing technique were used to increase the trustworthiness of their findings. Accordingly, the primary author maintained a positive relationship with participants during intervention follow-ups and established independence from caregiving facilities to minimize social desirability bias and bias views. To reduce bias and subjective interpretation, interviews were audio recorded, transcribed, and thoroughly evaluated, with participant quotes used to support the authors' findings".